# The Relationship between Family Characteristics and Adolescent Perception of the Quality of Family Communication

Martina Feric 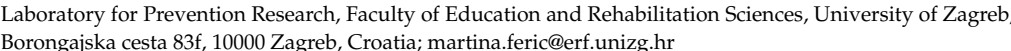

Laboratory for Prevention Research, Faculty of Education and Rehabilitation Sciences, University of Zagreb, Borongajska cesta 83f, 10000 Zagreb, Croatia; martina.feric@erf.unizg.hr

**Abstract:** Many studies consider family communication to be one of the most important protective factors for the positive development of children and adolescents in the family environment. This paper aims to clarify whether some characteristics of the family environment influence the quality of family communication in order to provide guidelines for the planning of prevention strategies that effectively improve the quality of family communication and, thus, the positive development of adolescents. Specifically, the aim is to investigate whether there are gender- and age-related differences in the assessment of the quality of family communication and whether there are differences in the assessment of the quality of family communication depending on some family characteristics. High school students from five large Croatian cities (Zagreb, Osijek, Split, Pula, and Varazdin) took part in this study. The quota sample is stratified by three Croatian high school programs, as well as by individual program orientations within each school. The results show that there are differences in assessment of the quality of family communication in relation to gender, age, living with both or one parent, and the educational and working status of the parents. The data suggest that, in addition to the timely implementation of evidence-based parenting and/or family-based prevention interventions, there is a need to invest in high-quality social policies that could lead to a better quality of family life by increasing the chances of higher educational attainment for (future) parents as well as adequate employment opportunities.

**Keywords:** adolescents; family communication; family characteristics

## 1. Introduction

The family, as the most important socialisation instance which has a strong influence on development, can strongly promote the achievement of positive outcomes for children and adolescents [1–3]. On the other hand, various characteristics of the family environment may represent risk factors for adolescents' involvement in risk behaviours, the progression of risk behaviours to conduct disorders, or risk factors for the development of mental health problems [4–6]. Over the years, many studies have examined various characteristics of the family environment and adolescent developmental outcomes, such as family structure (e.g., single parent families, divorced parents) (e.g., [7–11]), parenting styles (permissive, authoritative, authoritarian, uninvolved) (e.g., [12–15]), the quality of the parent–child relationship (e.g., [16–18]), parental monitoring (knowing where the child is, what friends they have, what they spend their money on) (e.g., [19–21]), and parental support (encouraging and giving physical affection) (e.g., [22–24]).

Communication in the family context is usually defined as the ability of family members to share their needs, feelings, and desires with each other and to respond positively to the changing needs of family members [25]. Family communication patterns emerge through the processes by which families create and share their social reality [26]. Although the quality of family communication is important throughout the lifespan of family/child development, it becomes crucial in adolescence, as this is the time when parents and adolescents face rapid developmental changes and must both adapt [27].

A review of the literature revealed a large number of studies on the influence of family communication on adolescent developmental outcomes (e.g., [9,28–32]). The quality of communication between family members contributes to the quality of the parent–child relationship, which, in turn, predicts children's well-being [33]. This is confirmed by numerous studies linking the quality of family communication to various adolescent developmental outcomes. The poor quality of family communication is associated with the development of internalized [34–36] and externalized problems in children and adolescents [34,37–39]. For example, adolescents who reported poorer communication with their parents also reported lower levels of parental support, which was positively related to the adolescents' depressive difficulties [34]. Similarly, difficulties in communicating with parents may increase anxiety in children [40]. The results of the study examining the characteristics of family relationships in families of children with externalizing behaviour problems [39] showed an association between externalizing behaviour problems in adolescents and poor family communication.

On the other hand, better communication between parents and adolescents and respectful communication between family members are associated with lower levels of unacceptable behaviours [34,41]. Research on adolescents at risk of developing mental health or behavioural problems has found that effective parent–child communication is a protective factor, while problematic communication between parents and children is a risk factor for poor psychosocial adjustment in adolescents [28,42]. For example, one study [43] showed a significant relationship between adolescent mental health and family communication patterns, including the quality of conversations between parents and adolescents, while another study [29] showed how the quality of family communication predicts adolescents' life satisfaction. The relationship between the quality of family communication and adolescents' life satisfaction has also been confirmed in many other studies (e.g., [9,44–46]). Research indicates that positive communication with parents is associated with greater life satisfaction and that this relationship is slightly greater for girls [44].

With regard to the assessment of family communication quality and the age of the study participants, one paper [47] presents the results of a longitudinal study showing that adolescents perceive lower family communication quality with increasing age. The study also shows that gender is an important variable in the assessment of the quality of family communication. It appears that girls rate the quality of the relationship with both parents higher than boys, which could also mean that girls rate the quality of family communication higher than boys [48–50]. Another author [51] notes that some studies show that girls have a stronger connection with their parents and share more information about their lives with them than boys, which may influence the evaluation of the quality of family communication. Although family communication may be related to the likelihood of young people engaging in risk behaviours and/or presenting poor mental health, research also shows that this likelihood differs for girls and boys for some risk behaviours. For example, it was found [52] that the quality of communication with parents was a protective factor for marijuana use and smoking in boys, whereas this relationship was not found in girls. As the issue of differences in both the assessment of the quality of family communication and the effects of family communication on gender-specific developmental outcomes is complex, further research is needed to gain a comprehensive understanding of this issue.

Family socioeconomic status (SES) is often interpreted as a combination of parents' educational level, parents' occupation, and family income [53] and is considered an important factor in predicting children's developmental outcomes [54]. In the last decade, there have been a few studies examining the relationship between family SES and child and adolescent developmental outcomes (e.g., [55–59]), but most of these studies were conducted more than two decades ago. However, the general influence of parental education level and parental work status on the quality of family communication is an almost-unexplored topic. Some studies investigated family SES and the quality of the parent–child relationship in said families, but these studies could not confirm the relationship (e.g., [60,61]).

The aim of this study is to investigate whether there are gender- and age-related differences in the assessment of the quality of family communication and whether there are

differences in the assessment of the quality of family communication depending on some family characteristics (living together with both or one parent, educational and working status of the parents). While there have been a number of studies (although not many in recent years) looking at differences in adolescents' perceptions of the quality of family community in relation to gender, age, and whether they live with both or only one parent, there have been almost no studies on differences in adolescents' perceptions of the quality of family community in relation to their parents' education and work status. One of the purposes of this study is to shed more light on these latter relationships.

## 2. Materials and Methods

This study was conducted in the period from October 2018 to May 2019 as part of the project "Positive Development of Adolescents in Croatia" (Laboratory for Prevention Research, Faculty of Education and Rehabilitation Sciences, University of Zagreb).

**Participants**. High school students from all grades of three- and four-year-long vocational schools and gymnasiums from five Croatian cities took part in this study: Zagreb, Split, Osijek, Pula, and Varazdin. In Zagreb, a sample of 15% of the total high school population was planned and stratified by three types of educational programs, in accordance with the data obtained from the Office of Education, Culture and Sports of the City of Zagreb. In the other cities, a quota of 25% of the total high school population was planned, but, in Pula and Split, 15% of the total high school population was reached. In the total sample, the number of students from each school is proportional to the ratio between the number of students in that school and the total population of high schools in the respective city. The total number of participants corresponds to students from a total of 77 schools from all five cities; in these schools, 47.5% of boys (N = 4.595), and 52.5% of girls (N = 5.087) took part in this study. No information on gender was available for 4.5% of the sample (N = 456). The distribution of study participants by age shows that the youngest (14 years, 3%) and the oldest (19 years, 1%) are the least represented, while the other age groups are relatively evenly distributed, especially between the ages of 15 and 17 years (15 years (29%), 16 years (26%), 17 years (25%), and 18 years (16%)). The average age was 16.2 years (SDage = 1.16). A total of 47.6% of the study participants came from the city of Zagreb, 16.8% from Osijek, 12.5% from Split, 16.2% from Varazdin, and 6.9% from Pula. Regarding of the educational program attended by the study participants, 50% of the study participants attended a four-year vocational program, 27% of them a three-year vocational program, while 23% of the participants attended a gymnasium program.

**Procedure**. The Ethics Committee of the Faculty of Education and Rehabilitation Sciences of the University of Zagreb and the National Agency for Education and Teacher Training have authorised the conduct of this study. In Croatia, research with children is conducted in accordance with the Code of Ethics for Research with Children [62], according to which children over 14 years of age give their independent consent (parental consent is not required). The participants were informed about the aims of the study and gave their written consent to participate in this study. Participation was voluntary and confidential. This study was conducted in groups using a questionnaire (paper-and-pencil method) during school hours by the researchers and trained graduate and undergraduate students of the Social Pedagogy programme (University of Zagreb, Faculty of Education and Rehabilitation Sciences).

**Instruments**. A battery of questionnaires with several scales was used in this study, and, for the purposes of this article, the data collected in the following scales were used:

(1) Demographic Data Questionnaire. The questionnaire contains questions to collect basic information about the study participants, such as gender, age, name of school, type of school attended, year of schooling, grade with which they completed the last school year, age of parents, with whom they live, parents' partner status, and their education and work status.

(2) Family Communication Questionnaire (modified from the Family Communication Scale, FACES-IV; [63]). The Family Communication Scale contains 10 items describing

the most important aspects of communication in the family system (on this study sample $\alpha$ = 0.94). The study participants circled the level of agreement with each item (e.g., "My family members know how to listen"; "The members of my family calmly solve problems") on a 5-point Likert scale ranging from (1) strongly disagree to (5) strongly agree. Higher scores mean better communication within the family.

**Data analysis**. Methods of descriptive statistics (frequencies, means, correlations) were used to describe the sample and the assessment of the quality of family communication. All statistical analyses were carried out using SPSS 21.0. Cronbach's alpha was calculated for the Family Communication Questionnaire, and the normality of the frequency distribution was tested using the Kolmogorov–Smirnov test. The test revealed a significant deviation of the frequency from a normal distribution (D(9887) = 0.045; $p$ = 0.000), which is why non-parametric methods of data processing, the Mann–Whitney and Kruskal–Wallis tests, were used to analyse the differences. The effect size for the Mann–Whitney test was calculated. When analysing the differences in all independent variables, the data from the Family Communication Questionnaire were recoded into three categories: (1) disagree/strongly disagree, (2) neither agree nor disagree, (3) agree/strongly agree. The differences between the individual variables (gender, age/years of schooling, living with one or both parents, parents' education, and parents' work status) were analysed using the average overall results of the Family Communication Questionnaire after the results had been recoded into the specified categories.

## 3. Results

### 3.1. Family Communication

The adolescents in our study assessed the quality of family communication using the FACES IV questionnaire. The frequencies and descriptive statistics are shown in Tables 1 and 2.

Looking at the mean values, the study participants rated all the variables with high values, i.e., they rated the quality of family communication as relatively high. The variables with the highest mean values were the following: "Family members are able to ask each other for what they want" (M = 4.13); "When family members ask each other questions, they get honest answers" (M = 4.10); "Family members are very good listeners" (M = 4.08); and "Family members try to understand each other's feelings" (M = 4.08). The variables with the lowest mean score were the following: "When angry, family members rarely say anything negative about each other" (M = 3.28); "Family members are satisfied with how they communicate with each other"; (M = 3.82) and "Family members can calmly discuss problems with each other" (M = 3.82).

**Table 1.** Family communication (%).

| Variables | 1 | 2 | 3 | 4 | 5 |
|---|---|---|---|---|---|
| Family members are satisfied with how they communicate with each other. | 4.1 | 9.1 | 17.9 | 36.7 | 31.0 |
| Family members are very good listeners. | 2.3 | 6.7 | 13.2 | 35.1 | 41.5 |
| Our family members show their affection for each other. | 2.6 | 5.3 | 16.8 | 35.2 | 38.5 |
| Family members are able to ask each other for what they want. | 1.9 | 4.4 | 14.4 | 36.4 | 41.5 |
| Family members can calmly discuss problems with each other. | 5.1 | 9.8 | 18.3 | 30.1 | 35.3 |
| Our family members discuss their ideas and beliefs with each other. | 3.7 | 7.2 | 17.3 | 33.4 | 37.1 |
| When family members ask each other questions, they get honest answers. | 2.3 | 5.5 | 15.4 | 32.8 | 42.8 |
| Family members try to understand each other's feelings. | 3.0 | 5.7 | 15.3 | 31.9 | 42.5 |
| When angry, family members rarely say anything negative about each other. | 12.8 | 15.7 | 23.8 | 24.2 | 22.2 |
| Our family members sincerely show their feelings to each other. | 3.6 | 6.6 | 18.0 | 30.0 | 40.5 |

Note: 1 = strongly disagree, 2 = disagree, 3 = neither agree nor disagree, 4 = agree, and 5 = strongly agree.

**Table 2.** Family communication—descriptive statistics.

| Variables | Min | Max | Mean | SD |
|---|---|---|---|---|
| Family members are satisfied with how they communicate with each other. | 1 | 5 | 3.82 | 1.097 |
| Family members are very good listeners. | 1 | 5 | 4.08 | 1.014 |
| Our family members show their affection for each other. | 1 | 5 | 4.03 | 1.006 |
| Family members are able to ask each other for what they want. | 1 | 5 | 4.13 | 0.950 |
| Family members can calmly discuss problems with each other. | 1 | 5 | 3.82 | 1.173 |
| Our family members discuss their ideas and beliefs with each other. | 1 | 5 | 3.94 | 1.085 |
| When family members ask each other questions, they get honest answers. | 1 | 5 | 4.10 | 1.084 |
| Family members try to understand each other's feelings. | 1 | 5 | 4.08 | 1.045 |
| When angry, family members rarely say anything negative about each other. | 1 | 5 | 3.28 | 1.321 |
| Our family members sincerely show their feelings to each other. | 1 | 5 | 3.99 | 1.090 |

Note: Min. = minimal value, Max = maximal value, Mean = mean value, and SD = standard deviation.

### 3.1.1. Gender Differences

The non-parametric Mann–Whitney test was used to test gender-specific differences in the assessment of the quality of family communication. The responses of all the participants who indicated their gender were included in the analysis. The results show that there is a significant difference in the assessment of the quality of family communication between boys and girls.

Table 3 shows that boys achieve a higher rank, and the Mann–Whitney test shows that they rate the quality of family communication higher than girls.

**Table 3.** Gender differences in the assessment of the quality of family communication.

| Gender | N | Min | Max | Mean | SD | Mean Rank | MWU | *p* | r |
|---|---|---|---|---|---|---|---|---|---|
| male | 4451 | 1 | 3 | 2.73 | 0.815 | 4944.88 | | | |
| female | 4993 | 1 | 3 | 2.60 | 0.925 | 4523.27 | 10,117,654.500 | 0.000 | 0.10 * |

Note: Min. = minimum value, Max = maximum value, Mean = mean value, SD = standard deviation, MWU = Mann–Whitney U test, *p* = significance, and r = effect size (* r < 0.3 = small effect).

### 3.1.2. Age Differences

Given the high correlation between the year of schooling and the age of the study participants (r = 0.893, *p* = 0.000) and the fact that the variables mentioned share 79.74% of the common variance, age differences in the assessment of the quality of family communication were tested using the year of schooling as an independent variable. The Kruskal–Wallis test was used to test for differences, and the Mann–Whitney test was used to test for differences between specific groups (year of schooling). The results of the Kruskal–Wallis (KW) test are shown in Table 4. The results of the analysis show that there is a significant difference in the assessment of the quality of family communication depending on the school year of the study participants ($\chi2$ = 76.027, df = 3, *p* = 0.000, Table 4). The assessment of the quality of family communication decreases the higher the school year of the study participants is. The Mann–Whitney test (MWU) shows that there are significant differences between the individual groups (Table 5).

**Table 4.** Age differences in the assessment of the quality of family communication (KW).

| Year of Schooling | Min | Max | Mean | SD | Mean Rank | $\chi2$ | df | *p* |
|---|---|---|---|---|---|---|---|---|
| 1st year of schooling | 1 | 3 | 2.74 | 0.550 | 5214.56 | | | 0.000 |
| 2nd year of schooling | 1 | 3 | 2.67 | 0.597 | 4952.30 | | | |
| 3rd year of schooling | 1 | 3 | 2.63 | 0.629 | 4828.28 | 76.027 | 3 | |
| 4th year of schooling | 1 | 3 | 2.60 | 0.643 | 4684.01 | | | |

Note: Min. = minimum value, Max = maximum value, Mean = mean value, SD = standard deviation, $\chi2$ = Chi-Square score, df = degrees of freedom, and *p* = significance.

**Table 5.** Age differences in the assessment of the quality of family communication (MWU).

| Year of Schooling | N | Mean Rank | MWU | p | r |
|---|---|---|---|---|---|
| 1st year of schooling | 2641 | 2682.98 | 3,232,647 | 0.000 | 0.06 * |
| 2nd year of schooling | 2586 | 2543.53 | | | |
| 1st year of schooling | 2641 | 2709.60 | 3,135,925 | 0.000 | 0.09 * |
| 3rd year of schooling | 2576 | 2505.86 | | | |
| 1st year of schooling | 2641 | 2463.99 | 2,432,394 | 0.000 | 0.12 * |
| 4th year of schooling | 2054 | 2210.99 | | | |
| 2nd year of schooling | 2586 | 2613.95 | 3,246,845 | 0.044 | 0.03 * |
| 3rdyear of schooling | 2576 | 2548.92 | | | |
| 2nd year of schooling | 2586 | 2381.79 | 2,523,175 | 0.000 | 0.06 * |
| 4th year of schooling | 2054 | 2254.97 | | | |
| 3rd year of schooling | 2576 | 2350.50 | 2,581,151 | 0.34 | 0.03 * |
| 4th year of schooling | 2054 | 2283.06 | | | |

Note: MWU = Mann–Whitney U-test, *p* = significance, and r = effect size (* r < 0.3 = small effect).

### 3.2. Family Characteristics

The differences in the assessment of the quality of family communication depending on the characteristics of the families in which the study participants live were analysed in relation to whether the adolescents live with one or both parents (living arrangements) as well as in relation to the parents' level of education and work status.

### 3.2.1. Living Arrangement

The majority of adolescents live with both parents, who are married, and, if cohabitation is included, 80.1% of the adolescents in our study live with both parents. A total of 13.5% of the adolescents have divorced parents, and only 0.7% of the adolescents do not live with either parent. In view of the significant deviation of the results of the questionnaire on family communication from a normal distribution, the Mann–Whitney test was carried out to examine whether there are differences in the assessment of the quality of family communication depending on whether a study participant lives with both parents or only with one parent. The complete data were collected from 9.817 study participants. The analysis shows that there are significant differences in the assessment of the quality of family communication when taking account whether the study participants live with both or only one parent. The results of the analysis are shown in Table 6.

**Table 6.** Differences in the assessment of the quality of family communication between participants living with both or only one parent.

| Living Arrangements | N | Min | Max | Mean | SD | Mean Rank | MWU | p | r |
|---|---|---|---|---|---|---|---|---|---|
| One parent | 1895 | 1 | 3 | 2.49 | 0.716 | 4369.15 | 6,483,079.000 | 0.000 | 0.121 * |
| Both parents | 7922 | 1 | 3 | 2.71 | 0.565 | 5038.14 | | | |

Note: Min. = minimum value, Max = maximum value, Mean = mean value, SD = standard deviation, MWU = Mann–Whitney U test, *p* = significance, and r = effect size (* r < 0.3 = small effect).

Table 6 shows that the study participants who live with both parents achieve higher ranks, and the results of the Mann–Whitney test show that the difference in ranks is significant. In other words, adolescents who live with both parents perceive the quality of family communication to be higher.

### 3.2.2. Parents' Education Level

The educational level of the parents was described in the questionnaire using five categories: incomplete elementary school, completed elementary school, completed secondary school, completed two years of higher education, and completed university education (bachelor's or master's degree) or more (e.g., PhD). To achieve greater consistency between the categories and make the results of the analysis more transparent, new cate-

gories of parental education level were created to examine differences in parental education levels: less than secondary education (fathers 5.3%; mothers 7.1%), high school education (fathers 59.0%; mothers 54.4%), and more than high school education (fathers 35.6%; mothers 38.5%). Although a large difference in frequencies between the individual groups is visible, the sample size allows these differences to be analysed [64]. The Kruskal–Wallis test (KW) was used to test for differences (Tables 7 and 8), and the Mann–Whitney test (MWU) was used to test for differences between the individual groups (Tables 9 and 10).

**Table 7.** Differences in the assessment of the quality of family communication in relation to fathers' education level (KW).

| Fathers' Level of Education | Min. | Max. | Mean | SD | Mean Rank | $\chi2$ | df | *p* |
|---|---|---|---|---|---|---|---|---|
| less than secondary education | 1 | 3 | 2.58 | 0.664 | 4532.67 | | | |
| high school education | 1 | 3 | 2.65 | 0.617 | 4804.70 | 17.968 | 2 | 0.000 |
| more than high school education | 1 | 3 | 2.67 | 0.572 | 4929.08 | | | |

Note: Min. = minimum value, Max = maximum value, Mean = mean value, SD = standard deviation, $\chi2$ = Chi-Square score, df = degrees of freedom, and *p* = significance.

**Table 8.** Differences in the assessment of the quality of family communication in relation to mothers' education level (KW).

| Mothers' Level of Education | Min. | Max. | Mean | SD | Mean Rank | $\chi2$ | df | *p* |
|---|---|---|---|---|---|---|---|---|
| less than secondary education | 1 | 3 | 2.56 | 0.694 | 4577.61 | | | |
| high school education | 1 | 3 | 2.66 | 0.612 | 4884.74 | 21.041 | 2 | 0.000 |
| more than high school education | 1 | 3 | 1.69 | 0.576 | 4986.08 | | | |

Note: Min. = minimum value, Max = maximum value, Mean = mean value, SD = standard deviation, $\chi2$ = Chi-Square score, df = degrees of freedom, and *p* = significance.

**Table 9.** Differences in the assessment of the quality of family communication in relation to fathers' education level (MWU).

| Fathers' Level of Education | N | Mean Rank | MWU | *p* | r |
|---|---|---|---|---|---|
| less than secondary education | 517 | 2951.13 | 1,391,830 | 0.007 | 0.03 * |
| high school education | 5704 | 3125.49 | | | |
| less than secondary education | 517 | 1840.54 | 817,658.5 | 0.000 | 0.06 * |
| more than high school education | 3447 | 2003.79 | | | |
| high school education | 5704 | 4532.71 | 9,578,219 | 0.007 | 0.03 * |
| more than high school education | 3447 | 4649.29 | | | |

Note: MWU = Mann–Whitney U-test, *p* = significance, and r = effect size (* r < 0.3 = small effect).

**Table 10.** Differences in the assessment of the quality of family communication in relation to mothers' education level (MWU).

| Mothers' Level of Education | N | Mean Rank | MWU | *p* | r |
|---|---|---|---|---|---|
| less than secondary education | 698 | 2842.95 | 1,740,428 | 0.001 | 0.04 * |
| high school education | 5320 | 3031.35 | | | |
| less than secondary education | 698 | 2084.17 | 1,210,797 | 0.000 | 0.06 * |
| more than high school education | 3785 | 2271.11 | | | |
| high school education | 5320 | 4513.89 | 9,860,042 | 0.028 | 0.02 * |
| more than high school education | 3785 | 4607.97 | | | |

Note: MWU = Mann–Whitney U-test, *p* = significance, and r = effect size (* r < 0.3 = small effect).

The results of the Kruskal–Wallis test show that there is a significant difference in the assessment of the quality of family communication in relation to the participants' fathers' level of education ($\chi2$ = 17.968, df = 2, *p* = 0.000) and the participants' mothers' level of education ($\chi2$ = 21.041, df = 2, *p* = 0.000). A look at the ranks shows that the study participants whose fathers have a higher level of education achieve higher ranks. The

Mann–Whitney test shows that there are significant differences in relation to the individual educational levels of the participants' fathers ($p \leq 0.05$). Similar results were obtained when analysing the differences in the perception of the quality of family communication in relation to the participants' mothers' education. The Mann–Whitney test confirms significant differences in relation to the mothers' individual levels of education ($p \leq 0.05$). Higher ranks are achieved by the participants whose mothers have completed a higher level of education. From the results obtained, it can be concluded that adolescents whose parents have a higher level of education perceive family communication to be of a higher quality.

### 3.2.3. Parents' Work Status

The differences in the assessment of the quality of family communication among the study participants were analysed with regard to permanent employment, occasional employment, parental unemployment, and retired parents. The majority of the participants' parents were permanently employed (76.2% of mothers and 80.7% of fathers). It should be emphasized that 13.2% of the mothers were unemployed, compared to 3.2% of the fathers. Given the specificity of each category of parental work status and their possible impact on family communication, all the categories were considered when analysing the differences, regardless of the significant differences in the representation of parents in each category. Given the size of the overall sample, this analysis is justified [64]. The Kruskal–Wallis test (KW) was used to test for differences (Tables 11 and 12), and the Mann–Whitney test (MWU) was used to test for differences between each group (Tables 13 and 14).

**Table 11.** Assessment of the quality of family communication in relation to the fathers' work status (KW).

| Fathers' Work Status | Min. | Max. | Mean | SD | Mean Rank | $\chi^2$ | df | *p* |
|---|---|---|---|---|---|---|---|---|
| permanently employed | 1 | 3 | 2.69 | 0.586 | 4790.77 | | | |
| occasional employment | 1 | 3 | 2.50 | 0.702 | 4166.50 | 64.685 | 3 | 0.000 |
| unemployed | 1 | 3 | 2.56 | 0.675 | 4341.44 | | | |
| retired | 1 | 3 | 2.59 | 0.659 | 4471.49 | | | |

Note: Min. = minimum value, Max = maximum value, Mean = mean value, SD = standard deviation, $\chi^2$ = Chi-Square score, df = degrees of freedom, and *p* = significance.

**Table 12.** Assessment of the quality of family communication in relation to the mothers' work status (KW).

| Mothers' Work Status | Min. | Max. | Mean | SD | Mean Rank | $\chi^2$ | df | *p* |
|---|---|---|---|---|---|---|---|---|
| permanently employed | 1 | 3 | 2.68 | 0.590 | 4840.36 | | | |
| occasional employment | 1 | 3 | 2.53 | 0.688 | 4326.36 | 46.102 | 3 | 0.000 |
| unemployed | 1 | 3 | 2.66 | 0.619 | 4771.18 | | | |
| retired | 1 | 3 | 2.51 | 0.692 | 4236.88 | | | |

Note: Min. = minimum value, Max = maximum value, Mean = mean value, SD = standard deviation, $\chi^2$ = Chi-Square score, df = degrees of freedom, and *p* = significance.

**Table 13.** Assessment of the quality of family communication in relation to the fathers' work status (MWU).

| Fathers' Work Status | N | Mean Rank | MWU | *p* | r |
|---|---|---|---|---|---|
| permanently employed | 7741 | 4142.79 | 1,611,723 | 0.000 | 0.07 * |
| occasional employment | 480 | 3598.26 | | | |
| permanently employed | 7741 | 4035.31 | 1,050,401 | 0.000 | 0.04 * |
| unemployed | 300 | 3651.84 | | | |
| permanently employed | 7741 | 4354.67 | 3,269,257 | 0.000 | 0.04 * |
| retired | 906 | 4061.95 | | | |
| occasional employment | 480 | 384.90 | 69,312.00 | 0.302 | 0.03 * |
| unemployed | 300 | 399.46 | | | |

**Table 13.** *Cont.*

| Fathers' Work Status | N | Mean Rank | MWU | *p* | r |
|---|---|---|---|---|---|
| occasional employment | 480 | 664.34 | 203,443.0 | 0.018 | 0.06 * |
| retired | 906 | 708.95 | | | |
| unemployed | 300 | 591.15 | 132,194.5 | 0.388 | 0.02 * |
| retired | 906 | 607.59 | | | |

Note: MWU = Mann–Whitney U-test, *p* = significance, and r = effect size (* r < 0.3 = small effect).

**Table 14.** Assessment of the quality of family communication in relation to the mothers' work status (MWU).

| Mothers' Work Status | N | Mean Rank | MWU | *p* | r |
|---|---|---|---|---|---|
| permanently employed | 7476 | 4094.01 | 2,141,814 | 0.000 | 0.07 * |
| occasional employment | 642 | 3675.66 | | | |
| permanently employed | 7476 | 4390.24 | 4,734,250 | 0.278 | 0.01 * |
| unemployed | 1285 | 4327.24 | | | |
| permanently employed | 7476 | 3833.11 | 548,667.0 | 0.000 | 0.04 * |
| retired | 168 | 3350.38 | | | |
| occasional employment | 642 | 904.69 | 374,409.0 | 0.000 | 0.09 * |
| unemployed | 1285 | 993.63 | | | |
| occasional employment | 642 | 407.01 | 52,959.00 | 0.672 | 0.01 * |
| retired | 168 | 399.73 | | | |
| unemployed | 1285 | 736.31 | 95,974.00 | 0.003 | 0.08 * |
| retired | 168 | 655.77 | | | |

Note: MWU = Mann–Whitney U-test, *p* = significance, and r = effect size (* r < 0.3 = small effect).

There are significant differences with regard to the work status of the fathers and the assessment of the quality of family communication by the study participants ($\chi 2 = 64.685$, df = 3, *p* = 0.000). Looking at the differences between the individual groups (work status), the Mann–Whitney test shows that there are significant differences in the assessment of the quality of family communication between the study participants whose fathers are permanently employed and all the other working statuses of the fathers (*p* = 0.000). The study participants whose fathers are permanently employed achieve significantly higher ranks than the participants whose fathers are in other employment statuses. There is also a significant difference in the assessment of the quality of family communication between the study participants whose fathers work occasionally and those whose fathers are retired (*p* = 0.018). Looking at the ranks, it can be seen that the participants whose fathers are retired achieve significantly higher ranks than the participants whose fathers are occasionally employed. There are no significant differences between the fathers' other work statuses and the participants' assessment of the quality of family communication (*p* ≥ 0.05). With regard to the working status of mothers and the participants' assessment of the quality of family communication, the analysis results also show that there are significant differences ($\chi 2 = 46.102$, df = 3, *p* = 0.000) with regard to the working status of mothers. The Mann–Whitney test shows that there are significant differences in the assessment of the quality of family communication between the participants whose mothers are permanently and occasionally employed and those whose mothers are retired (*p* ≤ 0.05) but not those participants whose mothers are unemployed (*p* ≥ 0.05). The participants whose mothers are permanently employed achieve significantly higher ranks than the participants whose mothers are occasionally employed or retired. There is also a significant difference in the assessment of the quality of family communication between the participants whose mothers are unemployed and those whose mothers are occasionally employed or retired. For example, the participants whose mothers are unemployed achieve significantly higher ranks than the participants whose mothers are occasionally employed or retired. From the data obtained, it can be

concluded that the participants whose mothers are permanently employed or unemployed rate the quality of family communication to be higher than the participants whose mothers are occasionally employed or retired.

## 4. Discussion

The aim of this study was to analyse the characteristics of the family environment of adolescents in Croatia and the quality of family communication. In terms of family characteristics, most of the adolescents in our study live with both parents, while 13.5% of them have divorced parents. Most of the parents in our study have a high school diploma, while slightly more of the mothers have completed higher education. The recent trend of more women than men completing higher levels of education is also confirmed by other sources (e.g., [65,66]). In terms of parental work status, most of the parents are in permanent employment. It is interesting to note that significantly more of the mothers than the fathers are unemployed, although more of the mothers have a university degree than the fathers. Unfortunately, this study did not collect any data on the reason for unemployment. One possible explanation for this is that a certain number of the mothers in our study is unemployed by choice, i.e., they have chosen to be "stay-at-home mums". There is a large difference seen between the number of mothers and fathers in our study who are retired, and the age of the fathers in the sample does not fully explain these data. It would certainly be worth investigating this in more detail.

The quality of family communication was perceived to be very high by the adolescents in our study. These results can be explained in two ways. On the one hand, Croatia is a very traditional country where the family is at the top of the value scale for many people and is somehow "guarded" [67]. On the other hand, it is possible that family in the Croatian context is still an environment rich in protective factors, i.e., it is perceived as such by adolescents [68,69]. The quality of family communication is assessed differently depending on the gender and age of adolescents. Girls perceive the quality of family communication to be significantly lower than boys. These data contradict other studies' findings that show that girls rate various characteristics of family communication higher than boys [49–52]. The explanation for such results could lie in the possibly greater relationship orientation and sensitivity of girls compared to boys [70] and, thus, in a more critical evaluation of the family environment, including the quality of family communication. The finding showing that the girls in our study perceive family communication to be of a lower quality than the boys is noteworthy, as the link between the quality of family communication and mental health problems is stronger in girls [32].

The assessment of the quality of family communication decreases with increasing age, which is consistent with other studies (e.g., [32,47]).

The results in our study show that family characteristics influence adolescents' assessment of the quality of family communication. Adolescents who live with both parents have a significantly higher perception of the quality of family communication. These results are consistent with the findings of other studies (e.g., [71,72]).

In addition, it has been shown that parents' level of education is one of the characteristics of the family context that influences adolescents' perception of family communication, i.e., the results of this study show that adolescents whose parents have a higher level of education perceive family communication to be qualitatively better. There are almost no studies in the literature that have investigated differences in the assessment of the quality of family communication depending on the parents' level of education. Studies that have looked at the influence of parental education on children's developmental outcomes show that parental education has an influence on children's developmental outcomes, especially the mother's education [73–75]. Some studies [76,77] have shown that the educational status of parents influences their parenting style. Based on these findings, it can be hypothesised that the educational level of parents may indirectly influence the quality of family communication (by influencing their parenting style). However, this hypothesis still needs to be verified by further studies.

The results of this study on the differences in the assessment of family communication in relation to parents' work status provide very interesting data. When it comes to the quality of family communication and the working status of fathers, the results show that adolescents whose fathers are permanently employed rate the quality of family communication to be higher than adolescents whose fathers have a different working status. However, there is also a significant difference in the assessment of the quality of family communication between adolescents whose fathers work occasionally and those whose fathers are retired. The results show that adolescents whose fathers are retired rate the quality of family communication higher than adolescents whose fathers work occasionally. Looking at the working status of mothers, the results show that adolescents whose mothers are permanently employed rate the quality of family communication higher than adolescents whose mothers are unemployed, occasionally employed or retired. At the same time, adolescent whose mothers are unemployed consider family communication to be of a higher quality than adolescents whose mothers are occasionally employed or retired. Like parents' level of education, the working status of parents in relation to the quality of family communication has also been little investigated. However, research shows that fathers' unemployment is associated with an assessment of a poorer quality of communication in unemployed fathers as opposed to employed fathers [78]. Research also shows that the influence of parental employment has an impact on child and adolescent developmental outcomes (e.g., [59,79]). More specifically, research confirms that parental employment is associated with fewer social and emotional difficulties in young people [79]. As it was shown in this study that a father's work status of "retirement" and a mother's status of "unemployment" are associated with higher ratings of the quality of family communication than other parental work statuses (with the exception of permanent employment), a question can be raised as to whether this is related to a higher amount of time that parents have available and could hypothetically invest in family life. Another hypothetical explanation that should be tested is that parents' current work status, if chosen by mothers or fathers, may contribute to a higher life satisfaction of the parents in question, which could consequently be reflected in the quality of family communication. The justification for such a hypothetical explanation lies in studies that link parents' life satisfaction to their employment status, among other factors (e.g., [80,81]).

The hypothetical explanations offered also point to the limitations of this study. In addition to the fact that our study's results are only based on statements by the adolescents queried about the quality of family communication (and not on statements by all family members), the limitations of this study also relate to the lack of data on the reason for the parents' work statuses. As mentioned above, it is possible that the results regarding the differences in the assessment of the quality of communication related to the parents' work status would be different if it were known whether the parents' work status was a choice or whether it was a matter of limited access to the labour market. Another limitation of the study is that it was conducted in urban areas (larger cities in the Republic of Croatia), meaning that the voice of adolescents living in rural areas is missing. For all these reasons, the results can only be generalized to all adolescents and their respective families located in the urban areas of Croatia, and generalization at the national level is only possible to a limited extent. Given these limitations, the recommendation for further research would certainly be to include adolescents from rural parts of Croatia so that the conclusions could be more comprehensive. Similarly, a clearer picture of the quality of communication in Croatian families would emerge if all family members were included in the assessment. Finally, given the results on the influence of parents' work status on the assessment of the quality of family communication, it would be important to further investigate the reasons for a parent's given work status in order to draw conclusions with greater scientific certainty, also because the explanation of these data is potentially important for the design of national social policies for families.

## 5. Conclusions

Adolescents in Croatia generally rate the quality of family communication to be high, but there are still differences in this assessment in relation to the age and gender of the adolescents, but also in relation to some family characteristics. The results of this study could provide evidence-based guidelines for effective prevention planning. For practitioners, the results of this study suggest that the implementation of effective universal family-based prevention interventions should begin in early adolescence at the latest. At the same time, the results indicate that universal prevention interventions must be gender-sensitive. In terms of selective prevention interventions, this study has identified groups of parents who need more support in developing skills for effective family communication. This study's findings, which point to differences in the perceptions of the quality of family communication in relation to parents' educational level and work status, could be relevant for decision/policy makers. These findings suggest that, in addition to implementing effective family-based interventions, national social policies need to be created to ensure access to education at all stages of life (e.g., the availability of lifelong learning) as well as access to the labour market. Investment in the timely implementation of evidence-based prevention interventions and the creation of evidence-based social policies would address the issue of creating an environment (family, society) that supports the positive development of young people from "top-down" and "bottom-up" perspectives or, in other words, comprehensively.

**Funding:** This research was funded via University Support Funding at the University of Zagreb, Croatia, with grant number PRAG 5501.

**Institutional Review Board Statement:** This study was conducted according to the guidelines of the Declaration of Helsinki and approved by Ethics Committee of the Faculty of Education and Rehabilitation Sciences, at the University of Zagreb (protocol code 251-74/17-01/2, 10 April 2017).

**Informed Consent Statement:** Informed consent was obtained from all the subjects involved in this study.

**Data Availability Statement:** The data presented in this study are available on request from the corresponding author. The data are not publicly available due to privacy issues.

**Acknowledgments:** I would like to thank the entire research team, the schools that took part in the study, and all the participants who shared their views with us.

**Conflicts of Interest:** The author declares no conflicts of interest.

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
