# Peer review of "The Relationship between Family Characteristics and Adolescent Perception of the Quality of Family Communication"

_adolescents, doi:10.3390/adolescents4010006_

Round 1
Reviewer 1 Report
Comments and Suggestions for Authors
Dear author,
It was with great pleasure and satisfaction that I reviewed this study. The theme and its characteristics make this work very interesting.
Overall, the formal and content aspects were achieved, and the type of sample collection is very relevant for external validity purposes. In order to improve the work presented, I make a set of considerations/suggestions:
Abstract: start with the objectives (their writing/articulation could be improved) and do not focus on the importance of the theme and the intended purpose.
I would remove the numeric values.
Introduction: Present relevant content to support the topic. However, it should indicate more studies that substantiate its premises (e.g., Lines 23-27, 28-31, 44-45).
Lines 28-31: these characteristics should be more specified.
Line 37 - review. Lots of replication of the same word (adolescence).
TO1. Page 2: the ideas are segmented and, since they refer to the consequences of communications failures, it could be important to relate them.
Lines 48-49; 60-64; 73-74: the ideas in the sentence are not well articulated.
Line 86 - pay attention to the parentheses.
Problematic: the objectives are in line with the study developed, however, their reformulation requires greater articulation and clarity.
Methodology: participants: indicate the range of ages.
Instruments: the way in which FACES results are interpreted is missing.
Analysis procedures: list all the techniques used (e.g., Cronbach's alpha, correlations are missing) and respective decision-making/interpretation. Also indicate the magnitudes of effect (here and its’ scores in the results’ chapter).
Results: at the beginning, a table with the average values, SD, range, % of the characteristics and FACES results would be very important, that is, a global characterization of the participants.
Notes must be indicated in the tables.
In table 2 they indicate classes. What do they refer to? Age classes or years of schooling? Make the text clearer.
Table 3: indicates Gender, which does not seem appropriate given the rest of the information.
It would be interesting to place graphs that help in the interpretation of the results found (e.g., line graphs), as well as the indication of M, SD, and Range in the tables.
Overall, the results are a little dense, especially because they are largely reported using the test used. I believe they should be refined, the name of the statistical tests removed and just indicate what was carried out (e.g., the comparisons).
Discussion: the results should be better articulated with previous studies (e.g., the fact that mothers have more education is already very common in research). Overall, I believe that it could be improved and the limitations more explored.
Conclusions: explore further the contribution of the study, added value/advantages and I recommend ending with an assumption in this sense.
Good work
Best regards
Author Response
Dear Reviewer,
Thank you for the effort and time you have invested in reviewing the paper. I appreciate it very much.
Following are the changes that were made based on the review:
|
recommendations |
modification |
|
Abstract: start with the objectives (their writing/articulation could be improved) and do not focus on the importance of the theme and the intended purpose. I would remove the numeric values. |
The summary has been rewritten and the numerical values have been deleted. |
|
Present relevant content to support the topic. However, it should indicate more studies that substantiate its premises |
23 new references have been added throughout the text
|
|
Lines 28-31: these characteristics should be more specified |
specified |
|
Page 2: the ideas are segmented and, since they refer to the consequences of communications failures, it could be important to relate them |
modified |
|
Lines 48-49; 60-64; 73-74: the ideas in the sentence are not well articulated.
|
modified |
|
Line 37 - review. Lots of replication of the same word (adolescence). |
modified |
|
Line 86 - pay attention to the parentheses.
|
corrected |
|
the objectives are in line with the study developed, however, their reformulation requires greater articulation and clarity. |
modified |
|
Methodology: participants: indicate the range of ages.
|
added |
|
Instruments: the way in which FACES results are interpreted is missing.
|
added |
|
Analysis procedures: list all the techniques used (e.g., Cronbach's alpha, correlations are missing) and respective decision-making/interpretation. Also indicate the magnitudes of effect (here and its’ scores in the results’ chapter). |
added |
|
Results: at the beginning, a table with the average values, SD, range, % of the characteristics and FACES results would be very important, that is, a global characterization of the participants. Notes must be indicated in the tables.
|
added |
|
In table 2 they indicate classes. What do they refer to? Age classes or years of schooling? Make the text clearer.
|
modified |
|
Table 3: indicates Gender, which does not seem appropriate given the rest of the information.
|
corrected |
|
It would be interesting to place graphs that help in the interpretation of the results found (e.g., line graphs), as well as the indication of M, SD, and Range in the tables.
|
M, SD and the Range were added in Tables |
|
Overall, the results are a little dense, especially because they are largely reported using the test used. I believe they should be refined, the name of the statistical tests removed and just indicate what was carried out (e.g., the comparisons).
|
modified |
|
Discussion: the results should be better articulated with previous studies (e.g., the fact that mothers have more education is already very common in research). Overall, I believe that it could be improved and the limitations more explored. |
modified |
|
Conclusions: explore further the contribution of the study, added value/advantages and I recommend ending with an assumption in this sense.
|
modified |
I hope that you will recognise all the changes as relevant and satisfactory in relation to the review.
Thank you once again.
Respectfully,
Author
Reviewer 2 Report
Comments and Suggestions for Authors
Title:
Family characteristics and adolescent perception of the quality of family communication
The reviewer’s comments
This manuscript is a worthy contribution to the investigation of the relationship between family characteristics and adolescent perception of the quality of family communication. However, the reviewer would like to see some revisions made to your manuscript.
1) Recommend the authors to think the title of manuscript.
The relationship between family characteristics and adolescent perception of the quality of family communication.
2) What new insight is your study offering to readers?
Strengthen this explanation in your introduction.
3) This study is based on the analysis of questionnaires applied to the different actors of such complex relations. Recommend the authors to think correctly the analysis of questionnaires is performed.
4) Please strengthen the conclusion and implications. Good finding suggestions for future practitioners and researchers.
5) This study is interesting. Reconsider after major revision.
Comments on the Quality of English LanguageTitle:
Family characteristics and adolescent perception of the quality of family communication
The reviewer’s comments
This manuscript is a worthy contribution to the investigation of the relationship between family characteristics and adolescent perception of the quality of family communication. However, the reviewer would like to see some revisions made to your manuscript.
1) Recommend the authors to think the title of manuscript.
The relationship between family characteristics and adolescent perception of the quality of family communication.
2) What new insight is your study offering to readers?
Strengthen this explanation in your introduction.
3) This study is based on the analysis of questionnaires applied to the different actors of such complex relations. Recommend the authors to think correctly the analysis of questionnaires is performed.
4) Please strengthen the conclusion and implications. Good finding suggestions for future practitioners and researchers.
5) This study is interesting. Reconsider after major revision.
Author Response
Dear Reviewer,
Thank you for the effort and time you have invested in reviewing the paper. I appreciate it very much.
Following are the changes that were made based on the review:
|
Recommend the authors to think the title of manuscript The relationship between family characteristics and adolescent perception of the quality of family communication |
Title is changed |
|
What new insight is your study offering to readers? Strengthen this explanation in your introduction. |
added |
|
This study is based on the analysis of questionnaires applied to the different actors of such complex relations. Recommend the authors to think correctly the analysis of questionnaires is performed. |
The presentation of the data (results) was changed at the request of another reviewer. I hope that this will lead to a “friendlier" understanding of the results. |
|
Please strengthen the conclusion and implications. Good finding suggestions for future practitioners and researchers. |
added |
I hope that you will recognise all the changes as relevant and satisfactory in relation to the review.
Thank you once again.
Respectfully,
Author
Reviewer 3 Report
Comments and Suggestions for Authors
This paper addresses the important topic of family communication. The topic is important with practical implication, but the authors need to make some revisions to this paper before it would be ready for publication.
INTRODUCTION
Why this study it is important? Please inform the readers.
MEASURES
Participants
Did the parents give the inform consent? The sample is constituted by adolescents.
Please add more information regarding Age, there is only information regarding the “The average
age was 16.2 years (SDage=1.16)”.
Procedure
Measures
Regarding The Family Communication Scale the α it is from the present study? Please clarify. Do you intend to say “grade”?
Data analysis
RESULTS
On line 183 the authors mentioned the term “class”. What do you mean by class? There is need a clarification. Perhaps “grade”?
Regarding Parent’s Education level there is need a clarification. What means “Tertiary professional education?” The sentence it is no clear form me. Based on the obtained
results, it can be concluded that adolescent whose parents are more educated rate family
communication as of higher quality. I don’t think the terms “more educated” it is the correct term.
On line 377 the sentence “in urban areas is missing” it is correct? Did you mean “rural”?
DISCUSSION
I would encourage authors to have an interpretation of the results that is more significant than what is currently here. For example: “The quality of family communication was perceived by adolescent as very high. These results can be explained in two ways. On the one hand, Croatia is a very traditional country where the family is at the top of the value scale for many people and is somehow "guarded". Secondly, it is possible that the family in the Croatian context is still an environment rich in protective factors, i.e. it is perceived as such by adolescents.” Please add references.
Regarding the follow result “It has been shown that the educational level of parents is one of the characteristics of the family context that affects family communication. The results of this study show that adolescents whose parents have a higher level of education perceive family communication as of higher quality.” Need clarification.
The next result needs a clarification based in references, there is lacking theoretical support. The results of the study on the differences in the assessment of family communication in relation to the parents' work status provide very interesting data. The is lacking references.
Add future studies suggestion.
Author Response
Dear Reviewer,
Thank you for the effort and time you have invested in reviewing the paper. I appreciate it very much.
Following are the changes that were made based on the review:
|
INTRODUCTION Why this study it is important? Please inform the readers. |
added |
|
MEASURES. Participants. Did the parents give the inform consent? The sample is constituted by adolescents |
Explanation added |
|
Please add more information regarding Age, there is only information regarding the “The average age was 16.2 years (SDage=1.16)”. |
Added |
|
Measures Regarding The Family Communication Scale the α it is from the present study? Please clarify. Do you intend to say “grade”? |
Clarified
"Grade" was replaced by "year of schooling" in the text at the request of another reviewer |
|
RESULTS On line 183 the authors mentioned the term “class”. What do you mean by class? There is need a clarification. Perhaps “grade”?
|
"Class" was replaced by "year of schooling" in the text at the request of another reviewer |
|
Data analysis Regarding Parent’s Education level there is need a clarification. What means “Tertiary professional education?” The sentence it is no clear form me. |
“tertiary professional education” has been replaced by “two years of higher education” |
|
Based on the obtained results, it can be concluded that adolescent whose parents are more educated rate familycommunication as of higher quality. I don’t think the terms “more educated” it is the correct term.
|
modified |
|
On line 377 the sentence “in urban areas is missing” it is correct? Did you mean “rural”?
|
corrected |
|
DISCUSSION I would encourage authors to have an interpretation of the results that is more significant than what is currently here. For example: “The quality of family communication was perceived by adolescent as very high. These results can be explained in two ways. On the one hand, Croatia is a very traditional country where the family is at the top of the value scale for many people and is somehow "guarded". Secondly, it is possible that the family in the Croatian context is still an environment rich in protective factors, i.e. it is perceived as such by adolescents.” Please add references. |
The text has been modifiy and new references added |
|
Regarding the follow result “It has been shown that the educational level of parents is one of the characteristics of the family context that affects family communication. The results of this study show that adolescents whose parents have a higher level of education perceive family communication as of higher quality.” Need clarification. |
Clarified
|
|
The next result needs a clarification based in references, there is lacking theoretical support. The results of the study on the differences in the assessment of family communication in relation to the parents' work status provide very interesting data. The is lacking references. |
23 new references have been added throughout the text
|
|
Add future studies suggestion. |
added |
I hope that you will recognise all the changes as relevant and satisfactory in relation to the review.
Thank you once again.
Respectfully,
Author
Round 2
Reviewer 2 Report
Comments and Suggestions for Authors
Title:
The relationship between family characteristics and adolescent perception of the quality of family communication
The reviewer’s comments
Thanks to the author for the correction. Revisions or explanations are all made according to the suggestions of the reviewers. Accept in present form.
Comments on the Quality of English Language
Title:
The relationship between family characteristics and adolescent perception of the quality of family communication
The reviewer’s comments
Thanks to the author for the correction. Revisions or explanations are all made according to the suggestions of the reviewers. Accept in present form.
Author Response
Dear Reviewer,
Thank you for your feedback. The English language has been slightly revised.
Respectfully,
Author
Reviewer 3 Report
Comments and Suggestions for Authors
Dear Authors thank you for attending my suggestions. I recommend to change the expressions "more educated". Please rearenge the sentence.
Author Response
Dear Reviewer,
Thank you for reading the article a second time, unfortunately I neglected to change the term “more educated” throughout the text.
Changes:
Sentence “Looking at the ranks it is evident that study participants whose fathers are more educated achieve higher ranks.” changed as follows “A look at the ranks shows that study participants whose fathers have a higher level of education achieve higher ranks.“
Sentences „Parents usually have a high school level of education, and mothers are slightly more likely to be more highly educated than fathers. The fact that women have recently become more educated than men is confirmed by other sources [e.g. 65, 66].“ changed as follows „Most parents have a high school diploma, while slightly more mothers have completed higher education. The recent trend of more women than men completing higher levels of education is also confirmed by other sources [e.g. 65, 66].“
Thank you again.
Respectfully,
Author